# Embedding Logical Queries on Knowledge Graphs

**William L. Hamilton**    **Payal Bajaj**    **Marinka Zitnik**    **Dan Jurafsky**[†]    **Jure Leskovec**

{wleif, pbajaj, jurafsky}@stanford.edu, {jure, marinka}@cs.stanford.edu
Stanford University, Department of Computer Science, [†]Department of Linguistics

## Abstract

Learning low-dimensional embeddings of knowledge graphs is a powerful approach used to predict unobserved or missing edges between entities. However, an open challenge in this area is developing techniques that can go beyond simple edge prediction and handle more complex logical queries, which might involve multiple unobserved edges, entities, and variables. For instance, given an incomplete biological knowledge graph, we might want to predict *what drugs are likely to target proteins involved with both diseases X and Y?*—a query that requires reasoning about all possible proteins that *might* interact with diseases X and Y. Here we introduce a framework to efficiently make predictions about conjunctive logical queries—a flexible but tractable subset of first-order logic—on incomplete knowledge graphs. In our approach, we embed graph nodes in a low-dimensional space and represent logical operators as learned geometric operations (e.g., translation, rotation) in this embedding space. By performing logical operations within a low-dimensional embedding space, our approach achieves a time complexity that is linear in the number of query variables, compared to the exponential complexity required by a naive enumeration-based approach. We demonstrate the utility of this framework in two application studies on real-world datasets with millions of relations: predicting logical relationships in a network of drug-gene-disease interactions and in a graph-based representation of social interactions derived from a popular web forum.

## 1 Introduction

A wide variety of heterogeneous data can be naturally represented as networks of interactions between typed entities, and a fundamental task in machine learning is developing techniques to discover or predict unobserved edges using this graph-structured data. Link prediction [25], recommender systems [48], and knowledge base completion [28] are all instances of this common task, where the goal is to predict unobserved edges between nodes in a graph using an observed set of training edges. However, an open challenge in this domain is developing techniques to make predictions about more complex graph queries that involve multiple unobserved edges, nodes, and even variables—rather than just single edges.

One particularly useful set of such graph queries, and the focus of this work, are *conjunctive queries*, which correspond to the subset of first-order logic using only the conjunction and existential quantification operators [1]. In terms of graph structure, conjunctive queries allow one to reason about the existence of subgraph relationships between sets of nodes, which makes conjunctive queries a natural focus for knowledge graph applications. For example, given an incomplete biological knowledge graph—containing known interactions between drugs, diseases, and proteins—one could pose the conjunctive query: "what protein nodes are likely to be associated with diseases that have both symptoms X and Y?" In this query, the disease node is an existentially quantified variable—i.e., we only care that *some* disease connects the protein node to these symptom nodes X and Y.

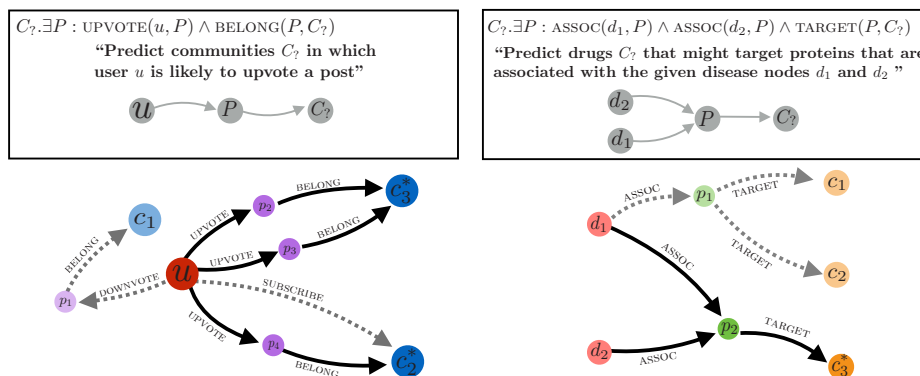

**Figure 1:** Two example conjunctive graph queries. In the boxes we show the query, its natural language interpretation, and the DAG that specifies this query's structure. Below these boxes we show subgraphs that satisfy the query (solid lines), but note that in practice, some of these edges might be missing, and we need to predict these missing edges in order for the query to be answered. Dashed lines denote edges that are irrelevant to the query. The example on the left shows a path query on the Reddit data; note that there are multiple nodes that satisfy this query, as well as multiple paths that reach the same node. The example on the right shows a more complex query with a polytree structure on the biological interaction data.

Valid answers to such a query correspond to subgraphs. However, since any edge in this biological interaction network might be unobserved, naively answering this query would require enumeration over all possible diseases.

In general, the query prediction task—where we want to predict likely answers to queries that can involve unobserved edges—is difficult because there are a combinatorial number of possible queries of interest, and any given conjunctive query can be satisfied by many (unobserved) subgraphs (Figure 1). For instance, a naive approach to make predictions about conjunctive queries would be the following: First, one would run an edge prediction model on all possible pairs of nodes, and—after obtaining these edge likelihoods—one would enumerate and score all candidate subgraphs that might satisfy a query. However, this naive enumeration approach could require computation time that is exponential in the number of existentially quantified (i.e., bound) variables in the query [12].

Here we address this challenge and develop graph query embeddings (GQEs), an embedding-based framework that can efficiently make predictions about conjunctive queries on incomplete knowledge graphs. The key idea behind GQEs is that we embed graph nodes in a low-dimensional space and represent logical operators as learned geometric operations (e.g., translation, rotation) in this embedding space. After training, we can use the model to predict which nodes are likely to satisfy any valid conjunctive query, even if the query involves unobserved edges. Moreover, we can make this prediction *efficiently*, in time complexity that is linear in the number of edges in the query and constant with respect to the size of the input network. We demonstrate the utility of GQEs in two application studies involving networks with millions of edges: discovering new interactions in a biomedical drug interaction network (e.g., "predict drugs that might treat diseases associated with protein X") and predicting social interactions on the website Reddit (e.g., "recommend posts that user A is likely to downvote, but user B is likely to upvote").

## 2 Related Work

Our framework builds upon a wealth of previous research at the intersection of embedding methods, knowledge graph completion, and logical reasoning.

**Logical reasoning and knowledge graphs**. Recent years have seen significant progress in using machine learning to reason with relational data [16], especially within the context of knowledge graph embeddings [6, 23, 18, 28, 29, 45], probabilistic soft logic [3], and differentiable tensor-based logic [11, 33]. However, existing work in this area primarily focuses on using logical reasoning to improve edge prediction in knowledge graphs [14, 13, 27], for example, by using logical rules as regularization [15, 20, 35, 37]. In contrast, we seek to directly make predictions about conjunctive logical queries. Another well-studied thread in this space involves leveraging knowledge graphs to improve natural language question answering (QA) [4, 5, 47]. However, the focus of these QA approaches is understanding natural language, whereas we focus on queries that are in logical form.

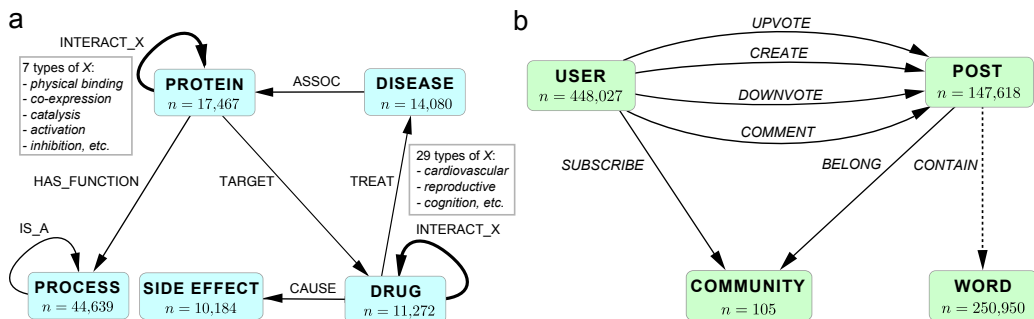

**Figure 2:** Schema diagrams for the biological interaction network and the Reddit data. Note that in the Reddit data words are only used as features for posts and are not used in any logical queries. Note also that for directed relationships, we add the inverses of these relationships to allow for a richer query space.

**Probabilistic databases**. Our research also draws inspiration from work on probabilistic databases [9, 12]. The primary distinction between our work and probabilistic databases is the following: Whereas probabilistic databases take a database containing probabilistic facts and score queries, we seek to predict *unobserved* logical relationships in a knowledge graph. Concretely, a distinguishing challenge in our setting is that while we are given a set of known edge relationships (i.e., facts), *all* missing edge relationships could possibly be true.

**Neural theorem proving**. Lastly, our work builds closely upon recent advancements in neural theorem proving [34, 43], which have demonstrated how neural networks can prove first-order logic statements in toy knowledge bases [36]. Our main contribution in this space is providing an efficient approach to embed a useful subset of first-order logic, demonstrating scalability to real-world network data with millions of edges.

## 3   Background and Preliminaries

We consider knowledge graphs (or heterogeneous networks) $\mathcal{G} = (\mathcal{V}, \mathcal{E})$ that consists of nodes $v \in \mathcal{V}$ and directed edges $e \in \mathcal{E}$ of various types. We will usually denote edges $e \in \mathcal{E}$ as binary predicates $e = \tau(u, v), \tau \in \mathcal{R}$, where $u, v \in \mathcal{V}$ are nodes with types $\gamma_1, \gamma_2, \in \Gamma$, respectively, and $\tau : \gamma_1 \times \gamma_2 \rightarrow \{\texttt{true}, \texttt{false}\}$ is the edge relation. When referring generically to nodes we use the letters $u$ and $v$ (with varying subscripts); however, in cases where type information is salient we will use distinct letters to denote nodes of different types (e.g., $d$ for a disease node in a biological network), and we omit subscripts whenever possible. Finally, we use lower-case script (e.g., $v_i$) for the actual graph nodes and upper-case script for variables whose domain is the set of graph nodes (e.g., $V_i$). Throughout this paper we use two real-world networks as running examples:

**Example 1: Drug interactions (Figure 2.a)**. A knowledge graph derived from a number from public biomedical databases (Appendix B). It consists of nodes corresponding to drugs, diseases, proteins, side effects, and biological processes. There are 42 different edge types, including multiple edge types between proteins (e.g., co-expression, binding interactions), edges denoting known drug-disease treatment pairs, and edges denoting experimentally documented side-effects of drugs. In total this dataset contains over 8 million edges between 97,000 nodes.

**Example 2: Reddit dynamics (Figure 2.b)**. We also consider a graph-based representation of Reddit, one of the most popular websites in the world. Reddit allows users to form topical communities, within which users can create and comment on posts (e.g., images, or links to news stories). We analyze all activity in 105 videogame related communities from May 1-5th, 2017 (Appendix B). In total this dataset contains over 4 million edges denoting interactions between users, communities and posts, with over 700,000 nodes in total (see Figure 2.b for the full schema). Edges exist to denote that a user created, "upvoted", or "downvoted" a post, as well as edges that indicate whether a user subscribes to a community

### 3.1 Conjunctive graph queries

In this work we seek to make predictions about *conjunctive graph queries* (Figure 1). Specifically, the queries $q \in \mathcal{Q}(\mathcal{G})$ that we consider can be written as:

$$q = V_? . \exists V_1, ..., V_m : e_1 \wedge e_2 \wedge ... \wedge e_n, \qquad (1)$$
$$\text{where } e_i = \tau(v_j, V_k), V_k \in \{V_?, V_1, ..., V_m\}, v_j \in \mathcal{V}, \tau \in \mathcal{R}$$
$$\text{or } e_i = \tau(V_j, V_k), V_j, V_k \in \{V_?, V_1, ..., V_m\}, j \neq k, \tau \in \mathcal{R}.$$

In Equation (1), $V_?$ denotes the *target variable* of the query, i.e., the node that we want the query to return, while $V_1, ..., V_m$ are existentially quantified *bound variable nodes*. The edges $e_i$ in the query can involve these variable nodes as well as *anchor nodes*, i.e., non-variable/constant nodes that form the input to the query, denoted in lower-case as $v_j$.

To give a concrete example using the biological interaction network (Figure 2.a), consider the query "return all drug nodes that are likely to target proteins that are associated with a given disease node $d$." We would write this query as:

$$q = C_?.\exists P : \text{ASSOC}(d, P) \wedge \text{TARGET}(P, C_?), \qquad (2)$$

and we say that the answer or *denotation* of this query $[\![q]\!]$ is the set of all drug nodes that are likely to be connected to node $d$ on a length-two path following edges that have types TARGET and ASSOC, respectively. Note that $d$ is an anchor node of the query: it is the input that we provide. In contrast, the upper-case nodes $C_?$ and $P$, are variables defined within the query, with the $P$ variable being existentially quantified. In terms of graph structure, Equation (2) corresponds to a path. Figure 1 contains a visual illustration of this idea.

Beyond paths, queries of the form in Equation (1) can also represent more complex relationships. For example, the query "return all drug nodes that are likely to target proteins that are associated with the given disease nodes $d_1$ and $d_2$" would be written as:

$$C_?.\exists P : \text{ASSOC}(d_1, P) \wedge \text{ASSOC}(d_2, P) \wedge \text{TARGET}(P, C_?).$$

In this query we have two anchor nodes $d_1$ and $d_2$, and the query corresponds to a polytree (Figure 1).

In general, we define the *dependency graph of a query* $q$ as the graph with edges $\mathcal{E}_q = \{e_1, ..., e_n\}$ formed between the anchor nodes $v_1, ..., v_k$ and the variable nodes $V_?, V_1, ..., V_m$ (Figure 1). For a query to be valid, its dependency graph must be a directed acyclic graph (DAG), with the anchor nodes as the source nodes of the DAG and the query target as the unique sink node. The DAG structure ensures that there are no contradictions or redundancies.

Note that there is an important distinction between the query DAG, which contains variables, and a subgraph structure in the knowledge graph that satisfies this query, i.e., a concrete assignment of the query variables (see Figure 1). For instance, it is possible for a query DAG to be satisfied by a subgraph that contains cycles, e.g., by having two bound variables evaluate to the same node.

**Observed vs. unobserved denotation sets**. If we view edge relations as binary predicates, the graph queries defined by Equation (1) correspond to a standard conjunctive query language [1], with the restriction that we allow at most one free variable. However, unlike standard queries on relational databases, we seek to discover or predict unobserved relationship and not just answer queries that exactly satisfy a set of observed edges. Formally, we assume that every query $q \in \mathcal{Q}(\mathcal{G})$ has some unobserved denotation set $[\![q]\!]$ that we are trying to predict, and we assume that $[\![q]\!]$ is not fully observed in our training data. To avoid confusion on this point, we also introduce the notion of the *observed denotation set* of a query, denoted $[\![q]\!]_{\text{train}}$, which corresponds to the set of nodes that exactly satisfy $q$ according to our observed, training edges. Thus, our goal is to train using example query-answer pairs that are known in the training data, i.e., $(q, v^*), v^* \in [\![q]\!]_{\text{train}}$, so that we can generalize to parts of the graph that involve missing edges, i.e., so that we can make predictions for query-answer pairs that rely on edges which are unobserved in the training data $(q, v^*), v^* \in [\![q]\!] \setminus [\![q]\!]_{\text{train}}$.

## 4 Proposed Approach

The key idea behind our approach is that we learn how to embed any conjunctive graph query into a low-dimensional space. This is achieved by representing logical query operations as geometric

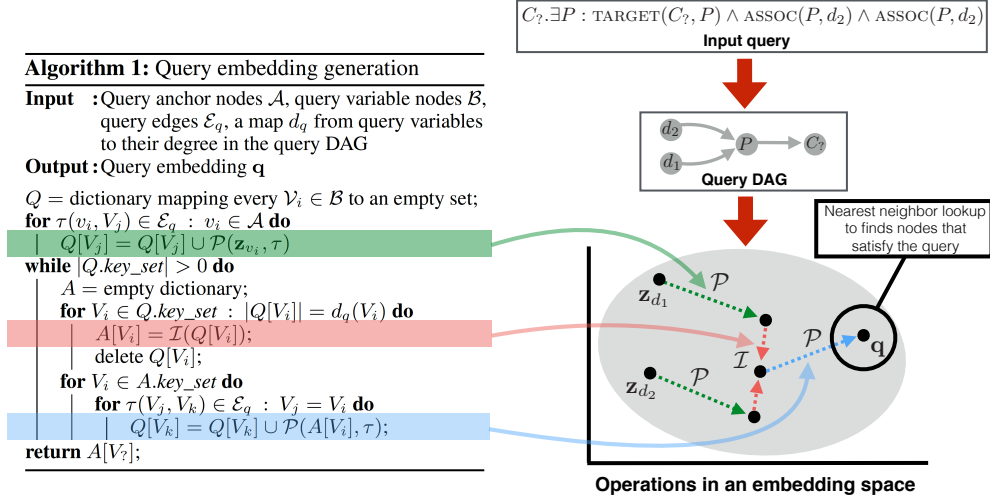

---

**Algorithm 1:** Query embedding generation

**Input** : Query anchor nodes $\mathcal{A}$, query variable nodes $\mathcal{B}$, query edges $\mathcal{E}_q$, a map $d_q$ from query variables to their degree in the query DAG

**Output** : Query embedding $\mathbf{q}$

$Q$ = dictionary mapping every $\mathcal{V}_i \in \mathcal{B}$ to an empty set;

**for** $\tau(v_i, V_j) \in \mathcal{E}_q \,:\, v_i \in \mathcal{A}$ **do**
  $\quad Q[V_j] = Q[V_j] \cup \mathcal{P}(\mathbf{z}_{v_i}, \tau)$

**while** $|Q.key\_set| > 0$ **do**
  $\quad A$ = empty dictionary;
  $\quad$ **for** $V_i \in Q.key\_set \,:\, |Q[V_i]| = d_q(V_i)$ **do**
    $\quad\quad A[V_i] = \mathcal{I}(Q[V_i]);$
    $\quad\quad$ delete $Q[V_i];$
  $\quad$ **for** $V_i \in A.key\_set$ **do**
    $\quad\quad$ **for** $\tau(V_j, V_k) \in \mathcal{E}_q \,:\, V_j = V_i$ **do**
      $\quad\quad\quad Q[V_k] = Q[V_k] \cup \mathcal{P}(A[V_i], \tau);$

**return** $A[V_?];$

**Figure 3:** Overview of GQE framework. Given an input query $q$, we represent this query according to its DAG structure, then we use Algorithm 1 to generate an embedding of the query based on this DAG. Algorithm 1 starts with the embeddings of the query's anchor nodes and iteratively applies geometric operations $\mathcal{P}$ and $\mathcal{I}$ to generate an embedding $\mathbf{q}$ that corresponds to the query. Finally, we can use the generated query embedding to predict the likelihood that a node satisfies the query, e.g., by nearest neighbor search in the embedding space.

operators that are jointly optimized on a low-dimensional embedding space along with a set of node embeddings. The core of our framework is Algorithm 1, which maps any conjunctive input query $q$ to an embedding $\mathbf{q} \in \mathbb{R}^d$ using two differentiable operators, $\mathcal{P}$ and $\mathcal{I}$, described below. The goal is to optimize these operators—along with embeddings for all graph nodes $\mathbf{z}_v \in \mathbb{R}^d, \forall v \in \mathcal{V}$—so that the embedding $\mathbf{q}$ for any query $q$ can be generated and used to predict the likelihood that a node $v$ satisfies the query $q$. In particular, we want to generate query embeddings $\mathbf{q}$ and node embeddings $\mathbf{z}_v$, so that the likelihood or "score" that $v \in [\![q]\!]$ is given by the distance between their respective embeddings:[1]

$$\texttt{score}(\mathbf{q}, \mathbf{z}_v) = \frac{\mathbf{q} \cdot \mathbf{z}_v}{\|\mathbf{q}\|\|\mathbf{z}_v\|}. \tag{3}$$

Thus, our goal is to generate an embedding $\mathbf{q}$ of a query that implicitly represents its denotation $[\![q]\!]$; i.e., we want to generate query embeddings so that $\texttt{score}(\mathbf{q}, \mathbf{z}_v) = 1, \forall v \in [\![q]\!]$ and $\texttt{score}(\mathbf{q}, \mathbf{z}_v) = 0, \forall v \notin [\![q]\!]$. At inference time, we take a query $q$, generate its corresponding embedding $\mathbf{q}$, and then perform nearest neighbor search—e.g., via efficient locality sensitive hashing [21]—in the embedding space to find nodes likely to satisfy this query (Figure 3).

To generate the embedding $\mathbf{q}$ for a query $q$ using Algorithm 1, we (i) represent the query using its DAG dependency graph, (ii) start with the embeddings $\mathbf{z}_{v_1}, ..., \mathbf{z}_{v_n}$ of its anchor nodes, and then (iii) we apply geometric operators, $\mathcal{P}$ and $\mathcal{I}$ (defined below) to these embeddings to obtain an embedding $\mathbf{q}$ of the query. In particular, we introduce two key geometric operators, both of which can be interpreted as manipulating the denotation set associated with a query in the embedding space.

**Geometric projection operator, $\mathcal{P}$:.** Given a query embedding $\mathbf{q}$ and an edge type $\tau$, the projection operator $\mathcal{P}$ outputs a new query embedding $\mathbf{q}' = \mathcal{P}(\mathbf{q}, \tau)$ whose corresponding denotation is $[\![q']\!] = \cup_{v \in [\![q]\!]} N(v, \tau)$, where $N(v, \tau)$ denotes the set of nodes connected to $v$ by edges of type $\tau$. Thus, $\mathcal{P}$ takes an embedding corresponding to a set of nodes $[\![q]\!]$ and produces a new embedding that corresponds to the union of all the neighbors of nodes in $[\![q]\!]$, by edges of type $\tau$. Following a long line of successful work on encoding edge and path relationships in knowledge graphs [23, 18, 28, 29], we implement $\mathcal{P}$ as follows:

$$\mathcal{P}(\mathbf{q}, \tau) = \mathbf{R}_\tau \mathbf{q}, \tag{4}$$

where $\mathbf{R}_\tau^{d \times d}$ is a trainable parameter matrix for edge type $\tau$. In the base case, if $\mathcal{P}$ is given a node embedding $\mathbf{z}_v$ and edge type $\tau$ as input, then it returns an embedding of the neighbor set $N(v, \tau)$.

**Geometric intersection operator, $\mathcal{I}$:.** Suppose we are given a set of query embeddings $\mathbf{q}_1, ..., \mathbf{q}_n$, all of which correspond to queries with the same output node type $\gamma$. The geometric intersection

operator $\mathcal{I}$ takes this set of query embeddings and produces a new embedding $\mathbf{q}'$ whose denotation corresponds to $[\![q']\!] = \cap_{i=1,...,n}[\![q]\!]_i$, i.e., it performs set intersection in the embedding space. While path projections of the form in Equation (4) have been considered in previous work on edge and path prediction, no previous work has considered such a geometric intersection operation. Motivated by recent advancements in deep learning on sets [32, 46], we implement $\mathcal{I}$ as:

$$\mathcal{I}(\{\mathbf{q}_1, ..., \mathbf{q}_n\}) = \mathbf{W}_\gamma \Psi \left( \mathrm{NN}_k(\mathbf{q}_i), \forall i = 1, ...n \right), \qquad (5)$$

where $\mathrm{NN}_k$ is a $k$-layer feedforward neural network, $\Psi$ is a symmetric vector function (e.g., an elementwise mean or min of a set over vectors), $\mathbf{W}_\gamma, \mathbf{B}_\gamma$ are trainable transformation matrices for each node type $\gamma \in \Gamma$, and ReLU denotes a rectified linear unit. In principle, any sufficiently expressive neural network that operates on sets could be also employed as the intersection operator (e.g., a variant of Equation 5 with more hidden layers), as long as this network is permutation invariant on its inputs [46].

**Query inference using $\mathcal{P}$ and $\mathcal{I}$.** Given the geometric projection operator $\mathcal{P}$ (Equation 4) and the geometric intersection operator $\mathcal{I}$ (Equation 5) we can use Algorithm 1 to efficiently generate an embedding $\mathbf{q}$ that corresponds to any DAG-structured conjunctive query $q$ on the network. To generate a query embedding, we start by projecting the anchor node embeddings according to their outgoing edges; then if a node has more than one incoming edge in the query DAG, we use the intersection operation to aggregate the incoming information, and we repeat this process as necessary until we reach the target variable of the query. In the end, Algorithm 1 generates an embedding $\mathbf{q}$ of a query in $O(d^2 E)$ operations, where $d$ is the embedding dimension and $E$ is the number of edges in the query DAG. Using the generated embedding $\mathbf{q}$ we can predict nodes that are likely to satisfy this query by doing a nearest neighbor search in the embedding space. Moreover, since the set of nodes is known in advance, this nearest neighbor search can be made highly efficient (i.e., sublinear in $|\mathcal{V}|$) using locality sensitive hashing, at a small approximation cost [21].

## 4.1 Theoretical analysis

Formally, we can show that in an ideal setting Algorithm 1 can exactly answer any conjunctive query on a network. This provides an equivalence between conjunctive queries on a network and sequences of geometric projection and intersection operations in an embedding space.

**Theorem 1.** *Given a network $\mathcal{G} = (\mathcal{V}, \mathcal{E})$, there exists a set of node embeddings $\mathbf{z}_v \in \mathbb{R}^d, \forall v \in \mathcal{V}$, geometric projection parameters $\mathbf{R}_\tau \in \mathbb{R}^{d \times d}, \forall \tau \in \mathcal{R}$, and geometric intersection parameters $\mathbf{W}_\gamma, \mathbf{B}_\gamma \in \mathbb{R}^{d \times d}, \forall \gamma \in \Gamma$ with $d = O(|V|)$ such that for all DAG-structured queries $q \in \mathcal{Q}(\mathcal{G})$ containing $E$ edges the following holds: Algorithm 1 can compute an embedding $\mathbf{q}$ of $q$ using $O(E)$ applications of the geometric operators $\mathcal{P}$ and $\mathcal{I}$ such that*

$$score(\mathbf{q}, \mathbf{z}_v) = \begin{cases} 0 & \text{if } v \notin [\![q]\!]_{train} \\ \alpha > 0 & \text{if } v \in [\![q]\!]_{train} \end{cases},$$

*i.e., the observed denotation set of the query $[\![q]\!]_{train}$ can be exactly computed in the embeddings space by Algorithm 1 using $O(E)$ applications of the geometric operators $\mathcal{P}$ and $\mathcal{I}$.*

Theorem 1 is a consequence of the correspondence between tensor algebra and logic [11] combined with the efficiency of DAG-structured conjunctive queries [1], and the full proof is in Appendix A.

## 4.2 Node embeddings

In principle any efficient differentiable algorithm that generates node embeddings can be used as the base of our query embeddings. Here we use a standard "bag-of-features" approach [44]. We assume that every node of type $\gamma$ has an associated binary feature vector $\mathbf{x}_u \in \mathbb{Z}^{m_\gamma}$, and we compute the node embedding as

$$\mathbf{z}_u = \frac{\mathbf{Z}_\gamma \mathbf{x}_u}{|\mathbf{x}_u|}, \qquad (6)$$

where $\mathbf{Z}_\gamma \in \mathbb{R}^{d \times m_\gamma}$ is a trainable embedding matrix. In our experiments, the $\mathbf{x}_u$ vectors are one-hot indicator vectors (e.g., each node gets its own embedding) except for posts in Reddit, where the features are binary indicators of what words occur in the post.

### 4.3 Other variants of our framework

Above we outlined one concrete implementation of our GQE framework. However, in principle, our framework can be implemented with alternative geometric projection $\mathcal{P}$ and intersection $\mathcal{I}$ operators. In particular, the projection operator can be implemented using any composable, embedding-based edge prediction model, as defined in Guu et al., 2015 [18]. For instance, we also consider variants of the geometric projection operator based on DistMult [45] and TransE [6]. In the DistMult model the matrices in Equation (4) are restricted to be diagonal, whereas in the TransE variant we replace Equation (4) with a translation operation, $\mathcal{P}_{\text{TransE}}(\mathbf{q}, \tau) = \mathbf{q} + \mathbf{r}_\tau$. Note, however, that our proof of Theorem 1 relies on specific properties of projection operator described in Equation (4).

### 4.4 Model training

The geometric projection operator $\mathcal{P}$, intersection operator $\mathcal{I}$, and node embedding parameters can be trained using stochastic gradient descent on a max-margin loss. To compute this loss given a training query $q$, we uniformly sample a positive example node $v^* \in [\![q]\!]_{\text{train}}$ and negative example node $v_N \notin [\![q]\!]_{\text{train}}$ from the training data and compute:

$$\mathcal{L}(q) = \max\left(0, 1 - \texttt{score}(\mathbf{q}, \mathbf{z}_{v^*}) + \texttt{score}(\mathbf{q}, \mathbf{z}_{v_N})\right).$$

For queries involving intersection operations, we use two types of negative samples: "standard" negative samples are randomly sampled from the subset of nodes that have the correct type for a query; in contrast, "hard" negative samples correspond to nodes that satisfy the query if a logical conjunction is relaxed to a disjunction. For example, for the query "return all drugs that are likely to treat disease $d_1$ and $d_2$", a hard negative example would be diseases that treat $d_1$ but not $d_2$.

## 5 Experiments

We run experiments on the biological interaction (Bio) and Reddit datasets (Figure 2). Code and data is available at `https://github.com/williamleif/graphqembed`.

### 5.1 Baselines and model variants

We consider variants of our framework using the projection operator in Equation 4 (termed Bilinear), as well as variants using TransE and DistMult as the projection operators (see Section 4.3). All variants use a single-layer neural network in Equation (5). As a baseline, we consider an enumeration approach that is trained end-to-end to perform edge prediction (using Bilinear, TransE, or DistMult) and scores possible subgraphs that could satisfy a query by taking the product (i.e., a soft-AND) of their individual edge likelihoods (using a sigmoid with a learned scaling factor to compute the edge likelihoods). However, this enumeration approach has exponential time complexity w.r.t. to the number of bound variables in a query and is intractable in many cases, so we only include it as a comparison point on the subset of queries with no bound variables. (A slightly less naive baseline variant where we simply use one-hot embeddings for nodes is similarly intractable due to having quadratic complexity w.r.t. to the number of nodes.) As additional ablations, we also consider simplified variants of our approach where we only train the projection operator $\mathcal{P}$ on edge prediction and where the intersection operator $\mathcal{I}$ is just an elementwise mean or min. This tests how well Algorithm 1 can answer conjunctive queries using standard node embeddings that are only trained to perform edge prediction. For all baselines and variants, we used PyTorch [30], the Adam optimizer, an embedding dimension $d = 128$, a batch size of 256, and tested learning rates $\{0.1, 0.01, 0.001\}$.

### 5.2 Dataset of train and test queries

To test our approach, we sample sets of train/test queries from a knowledge graph, i.e., pairs $(q, v^*)$, where $q$ is a query and $v^*$ is a node that satisfies this query. In our sampling scheme, we sample a fixed number of example queries for each possible query DAG structure (Figure 4, bottom). For each possible DAG structure, we sampled queries uniformly at random using a simple rejection sampling approach (described below).

To sample training queries, we first remove 10% of the edges uniformly at random from the graph and then perform sampling on this downsampled *training graph*. To sample test queries, we sample

**Table 1:** Performance on test queries for different variants of our framework. Results are macro-averaged across queries with different DAG structures (Figure 4, bottom). For queries involving intersections, we evaluate both using standard negative examples as well as "hard" negative examples (Section 4.4), giving both measures equal weight in the macro-average. Figure 4 breaks down the performance of the best model by query type.

|  |  | Bio data | | | Reddit data | | |
|---|---|---|---|---|---|---|---|
|  |  | Bilinear | DistMult | TransE | Bilinear | DistMult | TransE |
| GQE training | AUC | **91.0** | 90.7 | 88.7 | **76.4** | 73.3 | 75.9 |
| | APR | **91.5** | 91.3 | 89.9 | **78.7** | 74.7 | 78.4 |
| Edge training | AUC | 79.2 | 86.7 | 78.3 | 59.8 | 72.2 | 73.0 |
| | APR | 78.6 | 87.5 | 81.6 | 60.1 | 73.5 | 75.5 |

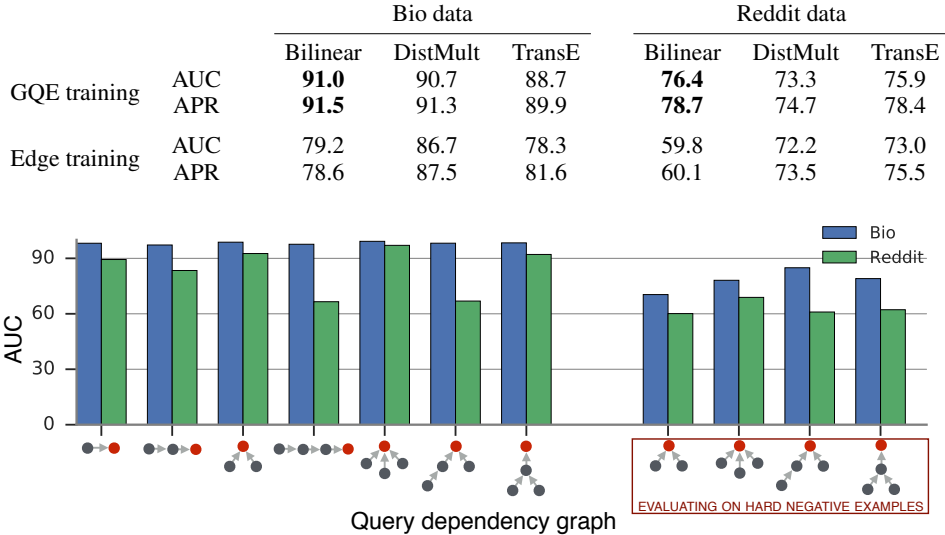

**Figure 4:** AUC of the Bilinear GQE model on both datasets, broken down according to test queries with different dependency graph structures, as well as test queries using standard or hard negative examples.

from the original graph (i.e., the complete graph without any removed edges), but we ensure that the test query examples are not directly answerable in the training graph. In other words, we ensure that every test query relies on at least one deleted edge (i.e., that for every test query example $(q, v^*)$, $v^* \notin [\![q]\!]_{\text{train}}$). This train/test setup ensures that a trivial baseline—which simply tries to answer a query by template matching on the observed training edges—will have an accuracy that is no better random guessing on the test set, i.e., that every test query can only be answered by inferring unobserved relationships.

**Sampling details**. In our sampling scheme, we sample a fixed number of example queries for each possible query DAG structure. In particular, given a DAG structure with $E$ edges—specified by a vector $\mathbf{d} = [d_1, d_2, ..., d_E]$ of node out degrees, which are sorted in topological order [42] —we sample edges using the following procedure: First we sample the query target node (i.e., the root of the DAG); next, we sample $d_1$ out-edges from this node and we add each of these sampled nodes to a queue; we then iteratively pop nodes from the queue, sampling $d_{i+1}$ neighbors from the $i$th node popped from the queue, and so on. If a node has $d_i = 0$, then this corresponds to an anchor node in the query. We use simple rejection sampling to cope with cases where the sampled nodes cannot satisfy a particular DAG structure, i.e., we repeatedly sample until we obtain $S$ example queries satisfying a particular query DAG structure.

**Training, validation, and test set details**. For training we sampled $10^6$ queries with two edges and $10^6$ queries with three edges, with equal numbers of samples for each different type of query DAG structure. For testing, we sampled 10,000 test queries for each DAG structure with two or three edges and ensured that these test queries involved missing edges (see above). We further sampled 1,000 test queries for each possible DAG structure to use for validation (e.g., for early stopping). We used all edges in the training graph as training examples for size-1 queries (i.e., edge prediction), and we used a $90/10$ split of the deleted edges to form the test and validation sets for size-1 queries.

## 5.3 Evaluation metrics

For a test query $q$ we evaluate how well the model ranks a node $v^*$ that does satisfy this query $v^* \in [\![q]\!]$ compared to negative example nodes that do not satisfy it, i.e., $v_N \notin [\![q]\!]$. We quantify this performance using the ROC AUC score and average percentile rank (APR). For the APR computation, we rank the true node against $\min(1000, |\{v \notin [\![q]\!]\}|)$ negative examples (that have the correct type

**Table 2:** Comparing GQE to an enumeration baseline that performs edge prediction and then computes logical conjunctions as products of edge likelihoods. AUC values are reported (with analogous results holding for the APR metric). Bio-H and Reddit-H denote evaluations where hard negative examples are used (see Section 5.3).

|  | Bio | Bio-H | Reddit | Reddit-H |
|---|---|---|---|---|
| Enum. Baseline | 0.985 | 0.731 | 0.910 | 0.643 |
| GQE | 0.989 | 0.743 | 0.948 | 0.645 |

for the query) and compute the percentile rank of the true node within this set. For queries containing intersections, we run both these metrics using both standard and "hard" negative examples to compute the ranking/classification scores, where "hard" negative examples are nodes that satisfy the query if a logical conjunction is relaxed to a disjunction.

## 5.4 Results and discussion

Table 1 contains the performance results for three variants of GQEs based on bilinear transformations (i.e., Equation 4), DistMult, and TransE, as well as the ablated models that are only trained on edge prediction (denoted Edge Training).[2] Overall, we can see that the full Bilinear model performs the best, with an AUC of 91.0 on the Bio data and an AUC of 76.4 on the Reddit data (macro-averaged across all query DAG structures of size 1-3). In Figure 4 we breakdown performance across different types of query dependency graph structures, and we can see that its performance on complex queries is very strong (relative to its performance on simple edge prediction), with long paths being the most difficult type of query.

Table 2 compares the best-performing GQE model to the best-performing enumeration-based baseline. The enumeration baseline is computationally intractable on queries with bound variables, so this comparison is restricted to the subset of queries with no bound variables. Even in this restricted setting, we see that GQE consistently outperforms the baseline. This demonstrates that performing logical operations in the embedding space is not only more efficient, it is also an effective alternative to enumerating the product of edge-likelihoods, even in cases where the latter is feasible.

**The importance of training on complex queries**. We found that explicitly training the model to predict complex queries was necessary to achieve strong performance (Table 1). Averaging across all model variants, we observed an average AUC improvement of 13.3% on the Bio data and 13.9% on the Reddit data (both $p < 0.001$, Wilcoxon signed-rank test) when using full GQE training compared to Edge Training. This shows that training on complex queries is a useful way to impose a meaningful logical structure on an embedding space and that optimizing for edge prediction alone does not necessarily lead to embeddings that are useful for more complex logical queries.

## 6    Conclusion

We proposed a framework to embed conjunctive graph queries, demonstrating how to map a practical subset of logic to efficient geometric operations in an embedding space. Our experiments showed that our approach can make accurate predictions on real-world data with millions of relations. Of course, there are limitations of our framework: for instance, it cannot handle logical negation or disjunction, and we also do not consider features on edges. Natural future directions include generalizing the space of logical queries—for example, by learning a geometric negation operator—and using graph neural networks [7, 17, 19] to incorporate richer feature information on nodes and edges.

**Acknowledgements**

The authors thank Alex Ratner, Stephen Bach, and Michele Catasta for their helpful discussions and comments on early drafts. This research has been supported in part by NSF IIS-1149837, DARPA SIMPLEX, Stanford Data Science Initiative, Huawei, and Chan Zuckerberg Biohub. WLH was also supported by the SAP Stanford Graduate Fellowship and an NSERC PGS-D grant.

## Footnotes

[1]We use the cosine distance, but in general other distance measures could be used.

[2]We selected the best $\Psi$ function and learning rate for each variant on the validation set.

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
