[Supplementary Material]

# Appendices to "Embedding Logical Queries on Knowledge Graphs"

**William L. Hamilton**     **Payal Bajaj**     **Marinka Zitnik**     **Dan Jurafsky**[†]     **Jure Leskovec**

{wleif, pbajaj, jurafsky}@stanford.edu, {jure, marinka}@cs.stanford.edu
Stanford University, Department of Computer Science, [†]Department of Linguistics

## Appendix A: Proof of Theorem 1

We restate Theorem 1 for completeness:

**Theorem 1.** *Given a network $\mathcal{G} = (\mathcal{V}, \mathcal{E})$, there exists a set of node embeddings $\mathbf{z}_v \in \mathbb{R}^d, \forall v \in \mathcal{V}$, geometric projection parameters $\mathbf{R}_\tau \in \mathbb{R}^{d \times d}, \forall \tau \in \mathcal{R}$, and geometric intersection parameters $\mathbf{W}_\gamma, \mathbf{B}_\gamma \in \mathbb{R}^{d \times d}, \forall \gamma \in \Gamma$ with $d = O(|V|)$ such that for all DAG-structured queries $q \in \mathcal{Q}(\mathcal{G})$ containing $E$ edges the following holds: Algorithm 1 can compute an embedding $\mathbf{q}$ of $q$ using $O(E)$ applications of the geometric operators $\mathcal{P}$ and $\mathcal{I}$ such that:*

$$score(\mathbf{q}, \mathbf{z}_v) = \begin{cases} 0 & \text{if } v \notin [\![q]\!]_{train} \\ \alpha > 0 & \text{if } v \in [\![q]\!]_{train} \end{cases},$$

*i.e., the observed denotation set of the query $[\![q]\!]_{train}$ can be exactly computed in the embeddings space by Algorithm 1 using $O(E)$ applications of the geometric operators $\mathcal{P}$ and $\mathcal{I}$.*

The proof of this theorem follows directly from two lemmas:

- Lemma 1 shows that any conjunctive query can be exactly represented in an embedding space of dimension $d = O(|\mathcal{V}|)$.
- Lemma 2 notes that Algorithm 1 terminates in $O(E)$ steps.

**Lemma 1.** *Given a network $\mathcal{G} = (\mathcal{V}, \mathcal{E})$, there exists a set of node embeddings $\mathbf{z}_v \in \mathbb{R}^d, \forall v \in \mathcal{V}$, geometric projection parameters $\mathbf{R}_\tau \in \mathbb{R}^{d \times d}, \forall \tau \in \mathcal{R}$, and geometric intersection parameters $\mathbf{W}_\gamma, \mathbf{B}_\gamma \in \mathbb{R}^{d \times d}, \forall \gamma \in \Gamma$ with $d = O(|V|)$ such that for any DAG-structured query $q \in \mathcal{Q}(\mathcal{G})$ an embedding $\mathbf{q}$ can be computed using $\mathcal{P}$ and $\mathcal{I}$ such that the following holds:*

$$score(\mathbf{q}, \mathbf{z}_v) = \begin{cases} 0 & \text{if } v \notin [\![q]\!]_{train} \\ \alpha > 0 & \text{if } v \in [\![q]\!]_{train} \end{cases},$$

*Proof.* Without loss of generality, we order all nodes by integer labels from $1...|V|$. Moreover, for simplicity, the subscript $i$ in our notation for a node $v_i$ will denote its index in this ordering. Next, we set the embedding for every node to be a one-hot indicator vector, i.e., $\mathbf{z}_{v_i}$ is a vector with all zeros except with a one at position $i$. Next, we set all the projection matrices $\mathbf{R}_\tau \in R^{|V| \times |V|}$ to be binary adjacency matrices, i.e., $\mathbf{R}_\tau(i,j) = 1$ iff $\tau(v_i, v_j) = \texttt{true}$. Finally, we set all the weight matrices in $\mathcal{I}$ to be the identity and set $\Psi = \min$, i.e., $\mathcal{I}$ is just an elementwise min over the input vectors.

Now, by Lemma 3 the denotation set $[\![q]\!]$ of a DAG-structured conjunctive query $q$ can be computed in a sequence $S$ of two kinds of set operations, applied to the initial input sets $\{v_1\}, ..., \{v_n\}$—where $v_1, ..., v_n$ are the anchor nodes of the query—and where the final output set is the query denotation:

- Set projections, with one defined for each edge type, $\tau$ and which map a set of nodes $\mathcal{S}$ to the set $\cup_{v_i \in \mathcal{S}} N(\tau, v_i)$.

- Set intersection (i.e., the basic set intersection operator) which takes a set of sets $\{\mathcal{S}_1, ..., \mathcal{S}_n\}$ and returns $\mathcal{S}_1 \cap, ..., \cap, \mathcal{S}_n$.

And we can easily show that $\mathcal{P}$ and $\mathcal{I}$ perform exactly these operations, when using the parameter settings outlined above, and we can complete our proof by induction. In particular, our inductive assumption is that sets $\mathcal{S}_i$ at step $k$ of the sequence $S$ are all represented as binary vectors $\mathbf{z}_{\mathcal{S}}$ with non-zeroes in the entries corresponding to the nodes in this set. Under this assumption, we have two cases, corresponding to what our next operation is in the sequence $S$:

1. If the next operation is a projection on a set $\mathcal{S}$ using edge relation $\tau$, then we can compute it as $\mathbf{R}_\tau \mathbf{z}_{\mathcal{S}}$, and by definition $\mathbf{R}_\tau \mathbf{z}_{\mathcal{S}}$ will have a non-zero entry at position $j$ iff there is at least one non-zero entry $i$ in $\mathbf{z}_{\mathcal{S}}$. That is, we will have that:

$$\texttt{score}(\mathbf{z}_u, \mathbf{R}_\tau \mathbf{z}_{\mathcal{S}}) = \begin{cases} 0 & \text{if } u \notin \cup_{v_i \in \mathcal{S}} N(\tau, v_i) \\ \alpha > 0 & \text{if } u \in \cup_{v_i \in \mathcal{S}} N(\tau, v_i). \end{cases}$$

2. If the next operation is an intersection of the set of sets $\{\mathcal{S}_1, ..., \mathcal{S}_n\}$, then we compute it as $\mathbf{z}' = \min\left(\{\mathbf{z}_{\mathcal{S}_1}, ..., \mathbf{z}_{\mathcal{S}_n}\}\right)$, and by definition $\mathbf{z}'$ will have non-zero entries only in positions where every one of the input vectors $\mathbf{z}_{\mathcal{S}_1}, ..., \mathbf{z}_{\mathcal{S}_n}$ has a non-zero. That is,

$$\texttt{score}(\mathbf{z}_{v_i}, \mathcal{I}(\{\mathbf{z}_{\mathcal{S}_1}, ..., \mathbf{z}_{\mathcal{S}_n}\})) = \begin{cases} 0 & \text{if } v_i \notin \mathcal{S}_1 \cap, ..., \cap, \mathcal{S}_n \\ \alpha > 0 & \text{if } v_i \in \mathcal{S}_1 \cap, ..., \cap, \mathcal{S}_n. \end{cases}$$

Finally, for the base case we have that the input anchor embeddings $\mathbf{z}_{v_1}, ..., \mathbf{z}_{v_n}$ represent the sets $\{v_1\}, ..., \{v_n\}$ by definition. $\qquad\square$

**Lemma 2.** *Algorithm 1 terminates in $O(E)$ operations, where $E$ is the number of edges in the query DAG.*

*Proof.* Algorithm 1 is identical to Kahn's algorithm for topologically sorting a DAG Kahn (1962), with the addition that we (i) apply $\mathcal{P}$ whenever we remove an edge from the DAG and (ii) run $\mathcal{I}$ whenever we pop a node from the queue. Thus, by direct analogy to Kahn's algorithm we require exactly $E$ applications of $\mathcal{P}$ and $V$ applications of $\mathcal{I}$, where $V$ is the number of nodes in the query DAG. Since $V$ is always less than $E$, we have $O(E)$ overall. $\qquad\square$

**Lemma 3.** *The denotation of any DAG-structured conjunctive query on a network can be obtained in a sequence of $O(E)$ applications of the following two operations:*

- *Set projections, with one defined for each edge type, $\tau$ and which map a set of nodes $\mathcal{S}$ to the set $\cup_{v_i \in \mathcal{S}} N(\tau, v_i)$.*

- *Set intersection (i.e., the basic set intersection operator) which takes a set of sets $\{\mathcal{S}_1, ..., \mathcal{S}_n\}$ and returns $\mathcal{S}_1 \cap, ..., \cap, \mathcal{S}_n$.*

*Proof.* Consider the two following simple cases:

1. For a query $C_? : \tau(v, C_?)$ the denotation is $N(v, \tau)$ by definition. This is simply a set projection.

2. For a query $C_? : \tau(v_1, C_?) \wedge \tau(v_2, C_?) \wedge ....\tau(v_n, C_?)$ the denotation is $\cap_{v_i \in \{v_1, ..., v_n\}} N(v_i, \tau)$ by definition. This is a sequence of $n$ set projections followed by a set intersection.

Now, suppose we process the query variables in a topological order and we perform induction on this ordering. Our inductive assumption is that after processing $k$ nodes in this ordering, for every variable $V_j, j \leq k$ in the query, we have a set $\mathcal{S}(V_j)$ of possible nodes that could be assigned to this variable.

Now, when we process the node $V_i$, we consider all of its incoming edges, and we have that:

$$\mathcal{S}(V_i) = \cap_{\tau_l(V_j, V_k) \in \mathcal{E}_q : V_k = V_i} \left(\cup_{v \in \mathcal{S}(V_j)} N(v, \tau)\right), \tag{1}$$

by definition. Moreover, by the inductive assumption the set $\mathcal{S}(V_j)$ for all nodes that have an outgoing edge to $V_i$ is known (because they must be earlier in the topological ordering). And Equation (1) requires only set projection and intersection operations, as defined above.

Finally, for the base case the set of possible assignments for the anchor nodes is given, and these nodes are guaranteed to be first in the DAG's topological ordering, by definition. $\qquad\square$

## Appendix B: Further dataset details

**Bio data**

The biological interaction network contains interactions between five types of biological entities (proteins, diseases, biological processes, side effects, and drugs). The network records 42 distinct types of biologically relevant molecular interactions between these entities, which we describe below.

Protein-protein interaction links describe relationships between proteins. We used the human protein-protein interaction (PPI) network compiled by Menche and others (2015) and Chatr-Aryamontri and others (2015), integrated with additional PPI information from Szklarczyk and others (2017), and Rolland and others (2014). The network contains physical interactions experimentally documented in humans, such as metabolic enzyme-coupled interactions and signaling interactions.

Drug-protein links describe the proteins targeted by a given drug. We obtained relationships between proteins and drugs from the STITCH database, which integrates various chemical and protein networks Szklarczyk and others (2015). Drug-drug interaction network contains 29 different types of edges (one for each type of polypharmacy side effects) and describes which drug pairs lead to which side effects Zitnik, Agrawal, and Leskovec (2018). We also pulled from databases detailing side effects (e.g., nausea, vomiting, headache, diarrhoea, and dermatitis) of individual drugs. The SIDER database contains drug-side effect associations Kuhn and others (2015) obtained by mining adverse events from drug label text. We integrated it with the OFFSIDES database, which details off-label associations between drugs and side effects Tatonetti and others (2012).

Disease-protein links describe proteins that, when mutated or otherwise genomically altered, lead to the development of a given disease. We obtained relationships between proteins and diseases from the DisGeNET database Piñero et al. (2015), which integrates data from expert-curated repositories. Drug-disease links describe diseases that a given drug treats. We used the RepoDB database Brown and Patel (2017) to obtain drug-disease links for all FDA-approved drugs in the U.S.

Finally, protein-process links describe biological processes (e.g., intracellular transport of molecules) that each protein is involved in. We obtained these links from the Gene Ontology database Ashburner et al. (2000) and we only considered experimentally verified links. Process-process links describe relationships between biological processes and were retrieved from the Gene Ontology graph.

We ignore in experiments any relation/edge-type with less than 1000 edges. Preprocessed versions of these datasets are publicly available at: `http://snap.stanford.edu/biodata/`.

**Reddit data**

Reddit is one of the largest websites in the world. As described in the main text we analyzed all activity (posts, comments, upvotes, downvotes, and user subscriptions) in 105 videogame related communities from May 1-5th, 2017. For the word features in the posts, we did not use a frequency threshold and included any word that occurs at least once in the data. We selected the subset of videogame communities by crawling the list of communities from the subreddit "/r/ListOfSubreddits", which contains volunteer curated lists of communities that have at least 50,000 subscribers. We selected all communities that were listed as being about specific videogames. All usernames were hashed prior to our analyses. This dataset cannot be made publicly available at this time.

## Appendix C: Further details on empirical evaluation

As noted in the main text, the code for our model is available at: `https://github.com/williamleif/graphqembed`

**Hyperparameter tuning**

As noted in the main text, we tested all models using different learning rates and symmetric vector aggregation functions $\Psi$, selecting the best performing model on the validation set. The other important hyperparameter for the methods is the embedding dimension $d$, which was set to $d = 128$ in all experiments. We chose $d = 128$ based upon early validation datasets on a subset of the Bio data. We tested embedding dimensions of 16, 64, 128, and 256; in these tests, we found performance increased until the dimension of 128 and then plateaued.

**Further training details**

During training of the full GQE framework, we first trained the model to convergence on edge prediction, and then trained on more complex queries, as we found that this led to better convergence. After training on edge prediction, in every batch of size $B$ we trained on $B$ queries of each type using standard negative samples and $B$ queries using hard negative samples. We weighted the contribution of path queries to the loss function with a factor of $0.01$ and intersection queries with a factor of $0.005$, as we found this was necessary to prevent exploding/diverging gradient estimates. We performed validation every 5000 batches to test for convergence. All of these settings were determined in early validation studies on a subset of the Bio data. Note that in order to effectively batch on the GPU, in every batch we only select queries that have the same edges/relations and DAG structure. This means that for some query types batches can be smaller than $B$ on occasion.

**Compute resources**

We trained the models on a server with 16 x Intel(R) Xeon(R) CPU E5-2623 v4 @ 2.60GHz processors, 512 GB RAM, and four NVIDIA Titan X Pascal GPUs with 12 GB of memory. This was a shared resource environment. Each model takes approximately 3 hours and three models could be concurrently run on a single GPU without significant slowdown. We expect all our experiments could be replicated in 48 hours or less on a single GPU, with sufficient RAM.

**Inverse edges**

Note that following Guu, Miller, and Liang (2015), we explicitly parameterize every edge as both the original edge and the inverse edge. For instance, if there is an edge TARGET$(u, v)$ in the network then we also add an edge TARGET$^{-1}(v, u)$ and the two relations TARGET and TARGET$^{-1}$ have separate parameterizations in the projection operator $\mathcal{P}$. This is necessary to obtain high performance on path queries because relations can be many-to-one and not necessarily perfect inverses. However, note also that whenever we remove an edge from the training set, we also remove the inverse edge, to prevent the existence of trivial test queries.