[Reviews · NeurIPS 2018]

Reviewer 1



This work proposes a method for answering complex conjunctive queries against an incomplete Knowledge Base. The method relies on node embeddings, and trainable differentiable representations for intersection and neighbourhood expansion operations, which can compute a prediction in linear time relative to the edges involved. The method is evaluated on a biomedical and a reddit dataset. The originality of this work lies in posing "soft" logical queries against a "hard" graph, rather than posing "hard" queries against a "soft" graph, as is done typically. The manuscript is clear, well-written and introduces notation using examples. The graphs and visualisations chosen throughout the manuscript are well-designed and help convey technical points. Careful thought has been put into selecting negatives during training, and there is an interesting ablation on different bilinear interaction functions. A major criticism of this work is that it is not compared against standard KB inference methods which operate on the individual links in a query independently. Of course standard KB inference methods cannot be used directly to answer complex queries, but a standard query language can, and in the simplest form it could just aggregate scores of all facts involved, independently scored by a standard KB inference method. The authors mention that "this would require computation time that is exponential in the number of edges" (line 41-42). However, in this simplest approach, a model could compute edge scores independently, which is linear in the number of query edges. This would at least reveal whether it is important to score edges in conjunction (the implicit underlying premise of this work), or whether it suffices to treat them independently at test time. A second criticism is that it is unclear whether the datasets chosen do actually test for a model's ability to answer logical conjunctions. 10% of edges are removed uniformly at random, and what exactly is required to answer a query is unclear. For example, in parallel query DAG structures, one branch alone might already suffice to unequivocally point to the correct answer. In fact, one observation from the experimental results is that linear chains are much harder to predict correctly than graphs with some sort of parallel structure. Furthermore, the longer the linear chain, the harder the task seems. This raises the question of redundancy in the dataset, and whether in those graphs with parallel structure one branch is already sufficient to specify the correct answer. Questions / Comments to the authors: - notation: line 80-81: double use of tau? - notation (Theorem 1, third line): What is V? - You mention different metrics: ROC AUC (section 5.1) and accuracy (Table 1) - training via sampling a fixed number of each query DAG structure was to facilitate batching? - line 272: where can this result be found? - is the macro-average across query DAG structures computed with adequate proportions, or uniformly weighted? - Can any general lessons be drawn from the selection of \Psi? - The origin of the drug interaction network dataset is not clear. How was this obtained?

Reviewer 2



The paper addresses the problem of answering complex queries in relational networks. It presents the query as a DAG. The proposed method learns to co-embed the data network and the query DAG in a common low dimensional vector space. The embedding of the query is recursively computed starting from constant nodes with a novel method. Finally, the solution to the query is found by the nearest neighbor in the embedding space. It then applies the method for queries on a drug interaction and social network. The paper addresses an important problem, namely combining logical reasoning and learning. While many papers consider query answering in a knowledge graph, the complicated form of query considered in this paper is novel and interesting and seems to be a step forward. Considering that knowledge graphs have been around for a long time, it is a curious case why expanding query complexity has not explored widely yet. I think, the motivations presented is a little weaker than reality and can be improved. The proposed solution is interesting too. Regarding the time complexity, looks like only the complexity of embedding is considered. However, after embedding the query responding it needs a nearest neighbor which is generally expensive. The naive solution proposed in lines 39-42 seems too weak. Looks like one easily can improve it based on efficient graph traversal. Also the link prediction phase is typically performed once, while prediction is repeated a lot. So making the first stage a little heavy might not be bad for most of real scenarios. The paper is well written and clear. The proposed projection method resembles [1, 2] below which are not cited. I do not see why the experimental results of the method is better than the edge training variant. I was expecting that the latter always give better results but be slower. My understanding is that the query is exactly answered in the edge training as opposed to being embedded and approximately answered as in the proposed method. Minor: The references page 10 is unfinished. It does not look like the network has to be heterogenous for the method to work. Overall, considering the strengths and shortcomings of the paper, I vote for an accept. -------------- [1] Paccanaro, Alberto, and Geoffrey E. Hinton. "Learning hierarchical structures with linear relational embedding.", in NIPS 2002. [2] Sutskever, Ilya, and Geoffrey E. Hinton. "Using matrices to model symbolic relationship", in NIPS 2009. --------------------- I read the rebuttal and found all the clarifications helpful to the reader. In particular, the burden of the nearest neighbor computation is confirmed and the paper has provided a solution for this challenge, which is good. I am happier with the submission after reading the rebuttal and having some discussions about it.

Reviewer 3



This paper proposes a system to query a knowledge graph combined with a knowledge base completion (KBC) system. Nodes (entities) and edge (relation) types are associated with suitable embeddings. The system described here can handle specific query structures. (The query is structured, not uncontrolled text.) Depending critically on the Deep Sets paper of Smola et al., this paper proposes compositional expressions to build up an embedding for a query in this class. Once the query is thus `compiled' into a suitable embedding vector, entities embedded to nearby vectors are presented as responses. Composing a query embedding involves definiting two operators in the query `language' and their denotations in the embedding domain. Given a set of nodes, the first operator (curiously called `projection') expands along edges of specified types to collect another set of nodes. The second operator (called `intersection') computes a query embedding from a set of nodes, with the underlying semantics of estimating the embedding of a node that is close to all the nodes. Although the query class is adequately motivated and the design decisions may be reasonable for building a practical system, I am not sure what the theoretical analysis means. Here are two specific issues. In equation (4), we get the impression that irrespective of how many nodes may be implicit in \mathbf{q}, it is represented by a vector of a fixed width/capacity d. This implies some form of symmetric aggregation to fixed capacity, which is troubling. For example, the centroid of a set of (node) vectors may be quite far from any node embedding. The problem persists when, in (4), we simply multiply this aggregate with a matrix and expect the resulting vector to continue to represent a different set of nodes (i.e. their embeddings). These operators can be cascaded, with potentially increasingly deleterious effects on the aggregate representation. In case of intersection, suppose we are looking for a vector v that is close to each of u_1, \ldots, u_m. Then, given a candidate v, we might create a network N(v, u_i) \in [0,1] that measures how close v and u_i are, and then we need to AND them, which would generally be expressed as a product of sigmoids, but perhaps min is OK. Again, for unclear reasons (except denotational efficiency), rather than judge the pairwise closeness first and then attempt a smooth AND, the paper retains (\mathbf{B}_\gamma transformed) node embeddings, does a symmetric aggregation, and only then applies a matrix (unclear why). Given the nonlinearities, these are not algebraically equivalent. It seems in both cases the authors are falling back heavily on the deep sets paper, but in practical applications, d = O(|V|) is rather unsatisfactory. If in reality you need d \ll O(|V|), you need to present intution about why this is possible, at least for the data sets you consider.